# Extracellular Signal-Regulated Kinases: One Pathway, Multiple Fates

**DOI:** 10.3390/cancers16010095

**Published:** 2023-12-24

**Authors:** Xavier Deschênes-Simard, Mohan Malleshaiah, Gerardo Ferbeyre

**Affiliations:** 1Montreal University Hospital Center (CHUM), Université de Montréal, Montréal, QC H3T 1J4, Canada; xavier.deschenes-simard@umontreal.ca; 2Department of Biochemistry and Molecular Medicine, Université de Montréal, Montréal, QC H3T 1J4, Canada; mohan.malleshaiah@ircm.qc.ca; 3Montreal Clinical Research Institute (IRCM), Montréal, QC H2W 1R7, Canada; 4Montreal Cancer Institute, CR-CHUM, Université de Montréal, Montréal, QC H3T 1J4, Canada

**Keywords:** cell proliferation, cell fate, ERK, senescence, cell signaling, EMT, apoptosis, pluripotency

## Abstract

**Simple Summary:**

The ERK pathway is a key cell signaling system activated by physiological factors or oncogenic mutations, influencing different cell outcomes. The protein kinases ERK1 and ERK2 (ERK) acting at the end of the pathway control whether cells grow, differentiate, age, or die. The exact mechanisms behind these outcomes are still not entirely clear, but they are affected by feedback signals and where ERK is activated in the cell. Generally, the outcome of ERK activation follows the Goldilocks principle—too much activity stops growth, while just the right amount supports it. Figuring out how this Goldilocks effect works involves looking at the stability and activity of ERK targets.

**Abstract:**

This comprehensive review delves into the multifaceted aspects of ERK signaling and the intricate mechanisms underlying distinct cellular fates. ERK1 and ERK2 (ERK) govern proliferation, transformation, epithelial–mesenchymal transition, differentiation, senescence, or cell death, contingent upon activation strength, duration, and context. The biochemical mechanisms underlying these outcomes are inadequately understood, shaped by signaling feedback and the spatial localization of ERK activation. Generally, ERK activation aligns with the Goldilocks principle in cell fate determination. Inadequate or excessive ERK activity hinders cell proliferation, while balanced activation promotes both cell proliferation and survival. Unraveling the intricacies of how the degree of ERK activation dictates cell fate requires deciphering mechanisms encompassing protein stability, transcription factors downstream of ERK, and the chromatin landscape.

## 1. Introduction

The extracellular signal-regulated kinases ERK1 and ERK2 (ERK) are the effectors of a signaling module activated by many membrane receptors to regulate cell fate (Figure 1). These kinases can control proliferation, transformation, epithelial–mesenchymal transition (EMT), differentiation, senescence, or cell death depending on the strength, duration, and context of their activation [1]. The biochemical mechanisms explaining how each different cell fate can be controlled by the same signaling module are poorly understood. ERK interacts with multiple proteins [2], many of which are modulated by ERK-dependent phosphorylation [3]. The biological outcome is contingent upon the specific ERK targets undergoing phosphorylation, which are determined by the dynamics and spatial localization of ERK activation. ERK dynamics is controlled by both positive and negative feedback loops (Figure 1) [4] while its location depends on its activation site (membrane vs. Golgi) [5,6] and the relocalization of activated ERK once they dissociate from their upstream activator complex [7]. Here, we concentrate on elucidating potential mechanisms that account for the diverse and occasionally contradictory cellular phenotypes resulting from ERK activation. The complex molecular intricacies governing the ERK pathway have been recently reviewed [8].

## 2. Proliferation and Survival

Cell proliferation was the first outcome linked to activation of the ERK pathway [9], and multiple ERK targets have been identified to explain this effect [3]. Initial studies on the response to ERK activation were conducted in tumor cell lines where proliferation is the default response due to their transformed state. Critical insights to understand how ERK controls proliferation came from studies of the rat adrenal pheochromocytoma cell lines (PC-12). In these cells, the epidermal growth factor (EGF) induces a transient activation of the pathway and cell proliferation, while the nerve growth factor (NGF) induces sustained activation and differentiation [10].

Given the association between proliferation and transient pathway activation, it logically follows that negative feedback mechanisms operate to prevent sustained activation. As expected, EGF induced these negative feedback mechanisms in PC12 cells. They include the phosphorylation of BRAF and MEK by ERK and the downregulation of SOS, a protein that stimulates the Ras GTPase in response to the activation of membrane receptors (Figure 1) [11,12]. In contrast, NGF induced positive feedback mechanisms that sustained the activation of the pathway [12]. The positive feedback was mediated through ERK-dependent RAF activation via protein kinase C (PKC) and inhibition of RAF kinase inhibitory protein (RKIP). Blocking these positive feedback mechanisms converted the NGF response from differentiation to proliferation [12]. Recent research has confirmed the antiproliferative effect of the PKCα-RAS-ERK positive feedback within the realm of intestinal epithelial cells. [13]. The need for negative feedback in the ERK pathway to promote proliferation has been observed as well in fibroblasts lacking expression of the dual specificity phosphatase DUSP4 [14] and in lung adenocarcinoma or leukemia cells treated with DUSP6 inhibitors [15,16]. In colorectal cancers, despite the presence of KRAS, NRAS and BRAF mutations in 50–60% of cancer patients [17], phospho-ERK staining was detectable in less than 10% of the cells [18]. This aligns with the necessity to prevent sustained elevation of ERK signaling for the maintenance of cell proliferation.

While in the PC12 system, proliferation is linked to a momentary activation of ERK, it’s important to note that time is a relative concept. A very brief pulse of ERK activation does not induce proliferation, but more prolonged stimulation does. Blenis and colleagues proposed a mechanism to explain why only long pulses of ERK activation stimulate cell proliferation. In their model, the initial ERK activation phosphorylates and stimulates transcription factors, leading to the expression of unstable immediate early genes (IEG). Some IEG products are then stabilized by ERK-catalyzed phosphorylation only if the ERK pathway remains activated [19]. In the Blenis model, signal duration is decoded in two steps, first transcriptional induction and then phosphorylation and stabilization of newly transcribed proteins (Figure 2). Several variations of this decoding mechanism have been described. First, sustained ERK activity can also stabilize mRNAs of late-response genes [20]. Second, multisite phosphorylation on the same target provides a mechanism to distinguish transient from lasting ERK activity. This was described for the phosphorylation of ELK-1 by ERK, where the more rapidly phosphorylated sites promote transcription by mediating interactions with the Mediator complex, but the late phosphorylation sites are inhibitory [21] (Figure 3). Finally, using live cell biosensors, Toetcher and colleagues established that IEG induction at the protein level depends on additional post-transcriptional regulatory mechanisms, which are gene-specific and are likely influenced by the signaling context of ERK activation [22]. 

Several transcription factors are activated by ERK-mediated phosphorylation (c-MYC, c-FOS, NR4A1, NR4A2, UBF, and MITF), and resistance to ERK pathway inhibitors can be mediated by reactivation of some of these transcription factors by ERK-independent pathways [23,24,25]. ERK mutations can also mediate resistance to pathway inhibitors [26]. These ERK mutants include *MAPK1^E322K^* (ERK2), present in head and neck carcinoma [27] and *MAPK3^R84S^* (ERK1), which can transform cells in culture [28]. In general, ERK mutants found in tumors exhibit a lower activity than endogenously activated ERK [28]. However, their role in cancer can be significant because proliferation requires a relatively low amplitude but lasting ERK activation [29]. Knowing that a relatively low ERK activation is sufficient to sustain cancer growth and survival is important for the clinical use of inhibitors of this pathway because they will need to be administered at doses that can inhibit more than 85% of the output [29]. In addition, pathway inhibitors release negative feedback signals conditioning cells for a rebound and drug resistance/addiction [30,31]. Of note, tumor cells that become addicted to pathway inhibitors proliferate with a very low level of ERK activation (only 2–3% of ERK was phosphorylated according to mass spectrometry) [31], indicating once more that anti-ERK therapies must inhibit the pathway with high efficiency to achieve an antiproliferative effect. 

In addition to its role in cell proliferation, ERK is required for cell survival in multiple cell types [32]. One important mechanism of cell survival triggered by ERK involves the phosphorylation of the pro-apoptotic protein BIM on Ser69, ultimately resulting in its ubiquitination and proteasomal degradation [33]. ERK also phosphorylates the transcription factor FoxO3, triggering its degradation, which results in inhibition of the expression of BIM [34]. The survival function of ERK could be targeted for anticancer therapies. Consistent with this idea, total ablation of ERK in mice completely blocked KRAS tumorigenesis. However, animals did not survive the loss of both ERK1 and ERK2 [35], suggesting potential toxicity issues with therapies that effectively eliminate ERK activity. 

### 2.1. ERK Localization in Cell Proliferation and Survival

The translocation of ERK from the cytoplasm to the nucleus after its activation is required for the stimulation of DNA synthesis and cell proliferation in response to growth factors [36]. PEA-15, a protein with a nuclear export sequence, binds to ERK, preventing its accumulation in the nucleus. Disabling PEA-15 increased cell proliferation, indicating again a critical role for ERK localization in deciding the outcome of its activation [37,38]. Conversely, preventing ERK nuclear translocation in melanoma cells decreased their survival even in cells that developed resistance to ERK pathway inhibitors [39,40].

Förster resonance energy transfer (FRET)-based ERK-biosensors can be targeted to different cellular locations to determine where ERK activity is confined after cellular stimulation. Intriguingly, the activation of the β2 adrenergic receptor, a G-protein-coupled receptor (GPCR), specifically activated ERK from endosomes and not from the plasma membrane. From endosomes, the signal propagates to the cytosol and the nucleus, stimulating MYC expression and cell proliferation [41]. It seems that GPCRs use a non-canonical pathway to stimulate ERK, warranting further studies into the comparative impact on cellular phenotype between signals originating from the endosomal pathway and those initiated by growth factor receptor tyrosine kinases at the plasma membrane.

### 2.2. ERK Pulses in Cell Proliferation and Survival

The pulsatile nature of several signaling pathways allows the regulation of different outcomes using the same signaling module. This occurs through the transmission of information by modulating the amplitude and frequency of the pulses (Figure 4). The proliferation fate in response to ERK requires frequency-modulated pulses of ERK activity as measured with a live cell ERK-reporter in cell culture [29] or a FRET (Förster resonance energy transfer)-ERK reporter in vivo [42]. In PC12 cells, where NGF usually triggers differentiation, administering NFG in a pulsatile manner can alter the outcome towards proliferation. Mathematical modeling has forecasted that pulsatile growth factor stimulation bypasses the positive feedback mechanisms responsible for driving cell differentiation [4].

Computer modeling suggested that negative feedback is central to pulse behavior in ERK signaling [29]. The precise mechanism responsible for decoding pulses of ERK activity was unidentified, but they seem to be translated into specific patterns of gene expression, suggesting that transcription factor activity is involved [22]. Work in yeast hints that gene activation can result from nonoverlapping pulses of an activator transcription factor (TF) and a repressor [43]. Oscillation in the DNA binding capacity of TFs may also help to dissociate them from the large number of binding sites present in the genome that effectively act as decoys [44]. This suggests that ERK pulses are decoded by affecting the dynamics of transcription factors regulating cell proliferation (Figure 4). 

Like proliferation, survival is causally linked to pulsatile ERK activation. High-frequency pulses characterize proliferation and survival in mammary acini grown in organoids, while low-frequency ERK pulses are associated with cell death and lumen formation. Inducing high-frequency ERK pulses promoted the survival of lumen cells. Interestingly, survival depended on ERK pulse frequency and not on the accumulation of ERK signals because spacing ERK pulses over longer times that achieve a similar overall ERK activity did not promote survival [45].

## 3. Differentiation

ERK operates downstream of growth factors that play a role in development, such as FGF (fibroblast growth factor), NGF (nerve growth factor), and many others [46,47]. Mouse models bearing conditional null alleles of ERK1 and ERK2 (Mapk3 and Mapk1) display craniofacial and cardiovascular defects. For the most part, the affected tissues involved derivatives of the neural crest, which constitutes a population of cells specified shortly after gastrulation. These neural crest cells give rise to multiple tissues, including bones, connective tissue, melanocytes, and various nerves [48,49,50]. In PC-12 cells, which originate from the neural crest, NGF treatment induces differentiation, and as discussed above, this requires sustained ERK activation and nuclear localization of ERK [2]. The ERK nuclear interactome in NGF-treated cells provided insights into how ERK regulates differentiation. Two critical targets are ERF, a suppressor of ETS transcription factor activity, and TRPS1, a suppressor of GATA transcription factors. Nuclear ERK phosphorylates ERF and TRPS1, inhibiting their activity and allowing both ETS- and GATA-mediated gene expression [2]. The data suggest that the regulation of cell differentiation by ERK depends on the integration of both duration and subcellular localization of ERK signaling. This integration provides more opportunities for regulation. Additional TFs regulating cell differentiation downstream of ERK in the PC-12 cell system were identified by studying the transcriptome after stimulation with either EGF or NFG. Short-term EGF stimulation-activated genes are regulated by E2F1, EBF1, SOX9, and SP1, while late-acting NGF-activated genes are regulated by BACH2, AP1, ETV4, and ELF2 [51]. The factors regulating the exchange of transcription factors between early and late time points are unknown. Late-acting TFs include the IEG stimulated during the initial wave of gene expression, as well as TFs induced by autocrine stimulation via secreted factors such as the urokinase-type plasminogen activator (uPA) and matrix metalloproteinases (MMPs) [51]. It is tempting to speculate that the fates controlled by sustained ERK activity depend on TFs, which do not contain inhibitory phosphorylation sites but only low-affinity, positively acting phosphorylation sites. In contrast, the proliferation fate will depend on factors such as ELK-1 (Figure 3), which contains both early-acting positive phosphorylation sites and late-acting negative phosphorylation sites (Figure 3).

The duration of ERK activity was also important to determine neurogenic endoderm (transient, 30 min) or gut ectoderm (sustained, 1 h) in flies. The role of signaling duration was demonstrated in this study by using optogenetics [52]. Interestingly, the mechanism specifying the gut ectoderm depended on cumulative ERK signaling, implying a memory of ERK activity across multiple cellular divisions [52] (Figure 5).

Finally, the activation of the ERK pathway by RAS can lead to either cell differentiation or apoptosis, depending on the origin of the RAS signals. Some studies suggest that RAS activation at the plasma membrane tends to induce cell differentiation, whereas signaling from the Golgi apparatus predominantly leads to apoptosis [6].

## 4. Pluripotency

ERK activity also regulates pluripotency. Active ERK1/2 inhibits the ‘naïve’ pluripotent state of mouse embryonic stem (ES) cells and favors their differentiation [55]. The naïve pluripotent state represents the pre-implantation pluripotent cells, which are unbiased for differentiation and therefore harbor unrestricted developmental potential [56]. Similarly, sustained ERK activity inhibits stem cells in colon crypt structures, while inhibiting ERK leads to the expansion of stem cells [57]. The combination of MEK and GSK3 inhibitors, which compose the medium formulation known as 2i, is used to maintain naïve stem cells in culture [58]. Several mechanisms inhibiting ERK in naïve pluripotent cells have been identified. Myc promotes the pluripotent cell fate by induction of the dual specificity phosphatases DUSP2 and 7, which can inhibit ERK activation [59]. The transcription factors PRDM14 and PRDM15 reinforce the naïve state by inhibiting FGF-ERK signaling and activating WNT signaling via R-spondin, mimicking the 2i condition [60]. ERK can also be inhibited via RSK-dependent negative feedback [61] and March5-mediated degradation of Prkar1a. The latter acts as a negative regulator of PKA, which in turn inhibits ERK by phosphorylation of RAF1 at Ser259 [62]. 

On the other hand, ERK inhibits the expression of pluripotency factors such as Nanog [63] via its ability to recruit PRC2 complexes and pause RNA polymerase II by phosphorylation at the CTD (C-terminal domain) at pluripotency genes [64]. ERK also regulates Nanog protein stability. In early embryos, cells experience a drop in ERK activity after mitosis. This drop leads to high levels of Nanog at the epiblast stage. The mechanism of ERK inactivation after mitosis is not fully understood, but it requires the E3 ubiquitin ligase APC (anaphase-promoting complex) [46]. In sharp contrast to its inhibitory role for mouse naïve ES cells, ERK1/2 activation through FGF promotes the ‘primed’ pluripotent epiblast stem cell state (EpiSC) [65]. The primed pluripotent state or EpiSCs represents the post-implantation pluripotent cells that are primed for developmental differentiation and therefore are more restricted in their developmental potential [56]. Similarly, ERK1/2 activation through FGF promotes the primed pluripotent state of human ES cells [66]. In this setting, ERK promotes the expression of genes essential for the survival and proliferation of human ES cells by binding to their promoters in association with the transcription factor ELK1 [67]. These works highlight the importance of ERK dynamics and chromatin interactions for developmental cell fate specification.

The ability of ERK to inhibit pluripotency may be relevant to cancer biology in some contexts. For example, in triple-negative breast cancer, chemotherapy resistance is associated with enrichment for cancer stem cells via reduction of ERK activity. In this case, the ERK pathway is attenuated at the level of MEK, which requires copper for its activity. Chemotherapy agents such as Carboplatin, Gemcitabine, and Paclitaxel induce copper chelation by elevating glutathione levels. Inhibition of MEK-ERK signaling reactivated FoxO3 nuclear translocation and transcriptional activation of the pluripotency factor Nanog [68]. Similarly, chemotherapy triggers DUSP9 through HIF activation, diminishing ERK signaling. This enables Nanog activation, thereby contributing to the enrichment of cancer stem cells [69].

## 5. Senescence

Cellular senescence in response to ERK pathway activation depends on high-intensity ERK signals [1,70,71]. In support of the link between intense ERK signals and tumor suppression, it has been shown that p53 contributes to ERK activation [72,73]. A model based on phosphorylation-dependent protein degradation was proposed based on evidence of multiple proteins degraded by the proteasome during oncogene-induced senescence [70]. The process was named SAPD (Senescence-Associated Protein Degradation). SAPD targets include several transcription factors, ribosome biogenesis factors, and mitochondrial proteins [74]. The SAPD model was further supported by the demonstration that inactivation of its targets (MYC, RSL1D1, and STAT3) is sufficient to induce senescence [70,75,76]. Intriguingly, ERK-dependent protein stabilization is linked to cell cycle progression and survival (Figure 2), while ERK-dependent protein degradation is linked to both inhibition of pluripotency and senescence (Figure 6). 

The activation of transcription factors is a key event to explain cell proliferation as well as senescence in response to ERK activation [19]. In one study, distinct genes were triggered following BRAFV600E activation in retinal epithelial cells, eliciting either proliferation or senescence based on the intensity of its activation: low levels prompted proliferation, whereas high levels induced senescence. Interestingly, gene expression changes relative to the control were calculated after considering both the time and the levels of ERK activation [77]. The data indicate that the relationship between ERK activation and cell proliferation is non-monotonic, fitting the Goldilocks principle where an intermediate level of ERK activity promotes proliferation, but higher levels block proliferation and trigger senescence (Figure 7). Intriguingly, during pancreatic tumorigenesis, a similar relationship between ERK activity and proliferation was documented. Proliferating malignant tumors have moderate levels of ERK activation, while benign lesions containing senescent cells have aberrantly high levels of activated ERK. In addition, increasing ERK activation in malignant pancreatic cancer cell lines triggered senescence and nucleolar alterations, notably the emergence of newly identified senescence-associated nucleolar foci [78]. Collectively, the recent data demonstrate that senescence triggered by prolonged or intense ERK activation entails distinct alterations in gene expression and nucleolar stress in addition to protein degradation (Figure 7).

The decision between senescence and transformation in oncogene-expressing cells is greatly influenced by factors that reinforce or attenuate ERK signaling. This model explains why, in some contexts, factors that reduce ERK signaling exhibit oncogenic properties while inhibiting them can induce senescence and/or impede tumor progression. These oncogenic ERK-pathway attenuators include dual specificity phosphatases [15,16,80,81,82,83], the tyrosine kinase FER [84], the S/T kinase Wnk2 [85], the miRNA binder Ago2 [86,87], the heat shock protein mortalin (HSPA9/GRP75/PBP74) [88] and the lysosomal cation transporter TRPML1 [89]. An intriguing attenuation mechanism in the ERK pathway was uncovered by measuring the abundances of non-phosphorylated, monophosphorylated (pT- or pY-), and double-phosphorylated (pTpY-) ERK1 and ERK2 by mass spectrometry. It’s noteworthy that solely double-phosphorylated ERK is functionally active. Tumor samples exhibited elevated levels of monophosphorylated pT-ERK1/2 in contrast to non-tumor samples, indicating a potential alteration in MEK-dependent ERK phosphorylation to prevent pathway hyperactivation [90]. Of note, some tumors retain high levels of ERK activation and still avoid senescence. This was discovered in colorectal cancer cells having KRAS G13D mutations that became resistant to MEKi. In this case, resistant cells develop an ERK-dependent epithelial–mesenchymal transition (EMT) that may inactivate senescence downstream of ERK [31].

## 6. EMT

ERK is an important activator of epithelial–mesenchymal transition (EMT) [91,92]. Several mechanisms have been proposed to explain how ERK activation triggers EMT. First, the kinase RPS6KA1 (Ribosomal Protein S6 Kinase A1), also known as RSK1, activates the late response transcription factor FRA1, which induces the expression of the EMT transcriptional repressors ZEB1 and ZEB2 [93,94]. Its paralog, RPS6KA3 (RSK2), also mediates EMT in response to activation of the tyrosine kinase receptor RON by macrophage-stimulating protein (MSP) in the context of cells having Ras mutations and activation of the ERK pathway [95]. Second, ERK can directly activate ZEB1, leading to the formation of a repressor complex between ZEB1 and the corepressor CtBP. The latter effectively suppresses the expression of E-cadherin, a hallmark event during the process of EMT [92,96]. Before ERK stimulation, ZEB1 is inactive in a complex with the MAPK Regulated Corepressor Interacting Protein-1 (MCRIP1). However, upon phosphorylation by ERK, MCRIP dissociates from ZEB1, allowing CtBP binding [96]. In agreement with this idea, the knockout of Zeb1 promotes cellular senescence by alleviating the repression of the CDK inhibitors p15INK4b and p21 [97]. In a lung cancer model, the initiation of Ras-dependent tumorigenesis required Zeb1 [98], which was likely due to the ability of the latter to block the antitumor senescence response triggered by Ras. Consistent with this hypothesis, in melanoma, high levels of ERK drive the expression of the senescence antagonist TWIST [99], leading to EMT [100]. These studies suggest that EMT regulators are critical components in cell fate determination downstream of ERK. 

The interplay between cellular senescence and EMT in response to ERK activation has been described in several experimental models of cancer. Early in pancreatic carcinogenesis, ERK cooperates with the TGFβ pathways to induce p21 and cell cycle arrest in benign cells. However, upon progression to pancreatic adenocarcinoma, ERK antagonizes TGFβ-induced cell cycle arrest and promotes EMT [101]. Both senescence [102,103,104] and EMT [105] are detected early during pancreatic carcinogenesis, suggesting EMT inducers may help elude senescence. In colorectal cancer, ERK activity is heterogeneous within tumors, and cells with the highest activity localize to the tumor edge. These high-ERK cells were present in KRAS mutated and wild-type cells and displayed decreased proliferation markers and signs of EMT.

Moreover, increasing ERK activity in colorectal cancer cells induced the EMT fate irrespective of their KRAS status. The tumor microenvironment likely regulates ERK levels in these tumors from a low basal level that promotes proliferation to a higher level that promotes EMT and cancer stem cell potential [106]. The senescence fate is likely evaded in these cases via the actions of the EMT mediators downstream of ERK (Figure 7). Since the expression of ERK targets depends on still poorly understood post-transcriptional controls [22], it is likely the EMT regulators could reprogram the response to ERK activation at this level. 

The context-dependent EMT regulation by the ERK pathway was revealed by studying the response to MEK inhibitors (MEKi) in colorectal cancer cells. Tumors that contain the BRAFV600E mutation develop resistance and addiction to MEKi via amplification of BRAFV600E. Drug withdrawal triggers senescence or apoptosis due to hyperactive ERK signaling in these addicted BRAFV600E-expressing cells. On the other hand, tumors with the mutation KRASG13D developed resistance due to ERK-dependent EMT [31]. A similar cell fate shift from ERK-dependent apoptosis to EMT was described in lung cancer cells that acquired resistance to the antifolate pemetrexed [107]. The chromatin landscape may provide the context in which ERK activation regulates EMT. As an example, targeting KRAS mutations to different skin epithelial cells in mice only generated mesenchymal lesions from hair follicle stem cells, which already possessed accessible chromatin at EMT transcription factors binding sites [108].

## 7. Apoptosis

Sustained ERK signaling can also trigger apoptosis, a fate observed in cells treated with different chemotherapeutic drugs or during neuronal cell death [109]. ERK activation also promotes cell death in response to low glucose by regulating GCN2/eIF2a/ATF4-dependent expression of pro-apoptotic molecules [110]. The apoptosis fate in response to ERK activation also depends on reaching a threshold of high ERK activity as reported for senescence [79] (Figure 7. Interestingly, cell death in response to supraphysiological ERK activation was partially dependent on secreted factors [111]. Of interest is that apoptosis induced by high ERK activity upon inhibition of DUSP6 was exploited to obtain a therapeutic response against chronic lymphocytic leukemia [112].

## 8. Conclusions

Cell fate in response to activated ERK pathway is context-dependent and influenced by the location and the dynamics of ERK activation: sustained vs. pulsatile. The role of negative and positive feedback mechanisms as drivers of ERK dynamics has been captured in several mathematical models, but the molecular details of the pathway are still poorly understood. We do not know to what extent growth factor stimulation also follows pulsatile behaviors in vivo, although it is clear that artificially imposed growth factor pulses can change cell fate [4]. The intensity and frequency of the ERK pulses convey information decoded by transcription factors, chromatin, and perhaps other effector molecules. In addition, the final protein output of ERK-target genes depends on post-transcriptional controls that are still largely unexplored [22]. The utilization of heightened ERK signaling for therapeutic purposes should incorporate a strategy to prevent epithelial–mesenchymal transition (EMT). Conversely, approaches aiming to reduce ERK signaling should steer clear of promoting an enrichment of tumor stem cells [68,113]. The interpretation of ERK pathway activation in tumors could be enhanced by developing context-specific biomarkers that are linked to each possible outcome associated with ERK inhibition or activation.

## Figures and Tables

**Figure 1 cancers-16-00095-f001:**
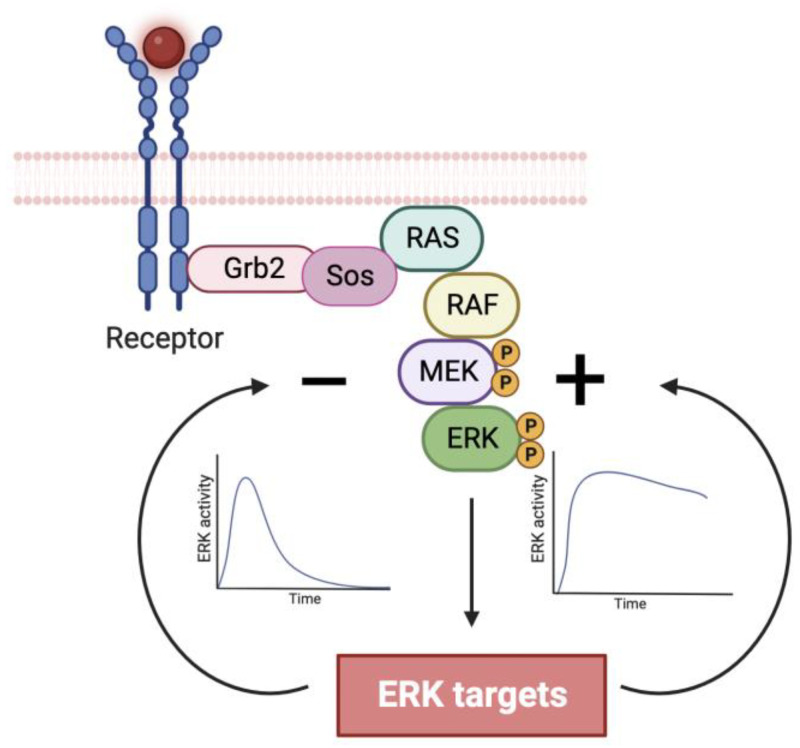
Simplified schematic of the activation of the ERK pathway by ligands binding to membrane receptors. Receptors activate the GTPase RAS via Grb2 (Growth Factor Receptor–Bound Protein 2) and Sos (Son of Sevenless). The signaling module downstream of RAS includes a cascade of kinases formed by RAF, MEK, and ERK. The outcomes of the pathway depend largely on the phosphorylation of ERK targets that modulate protein activity and gene expression but also attenuate (negative feedback) or reinforce the pathway (positive feedback). (Adapted from BioRender).

**Figure 2 cancers-16-00095-f002:**
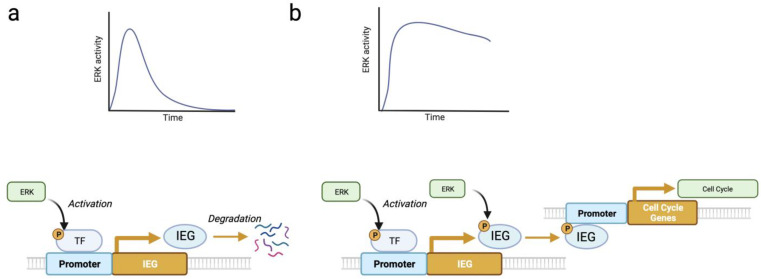
Decoding ERK-signaling duration by integrating ERK-dependent immediate early genes (IEGs) transcription (early effect) with ERK-dependent IEG products stabilization (late effect). (**a**) IEG are induced by transcription factors phosphorylated by ERK, but they are subsequently degraded. (**b**) Sustained ERK activity stabilizes IEG by further phosphorylation events. These IEGs code for transcription factors that control cell cycle genes. This model was proposed by Blenis and colleagues [19]. Long-lasting ERK signals can also stabilize mRNA of late response genes [20], although the direct targets of ERK involved were not identified (adapted from BioRender).

**Figure 3 cancers-16-00095-f003:**
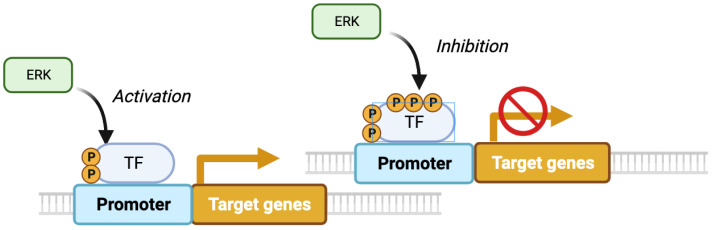
Decoding ERK-signaling duration and intensity by multisite phosphorylation. In this model, originally described for Elk1 phosphorylation, prolonged ERK signals are required to phosphorylate low-affinity inhibitory sites that turn off early-acting transcription factors [21] (adapted from BioRender).

**Figure 4 cancers-16-00095-f004:**
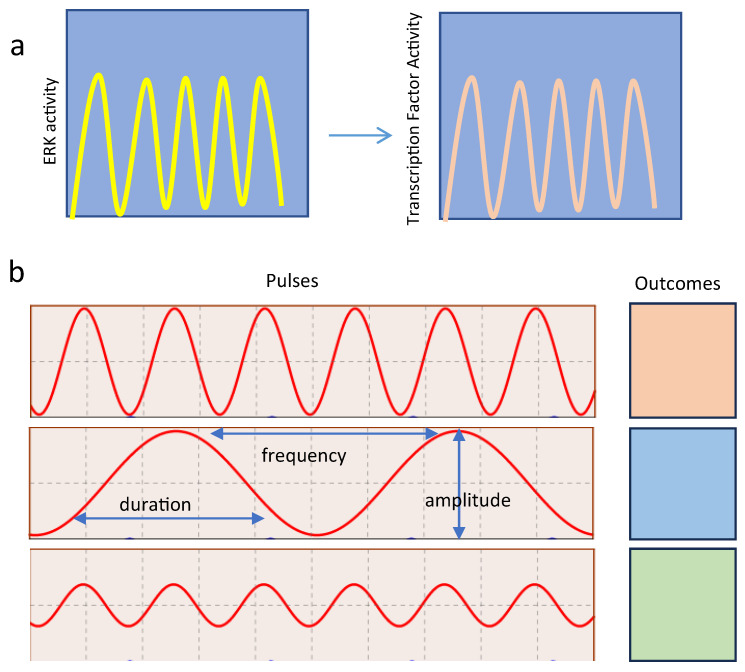
Decoding ERK oscillations by pulses of transcription factor activation. (**a**) Pulses of ERK activities detected by live-cell reporters [4,29,42] could be transmitted to effector molecules such as transcription factors [22]. (**b**) Modulating the frequency (# oscillation/time), amplitude and duration (the time it takes for a single oscillation) of ERK activity pulses can control different outcomes [4].

**Figure 5 cancers-16-00095-f005:**
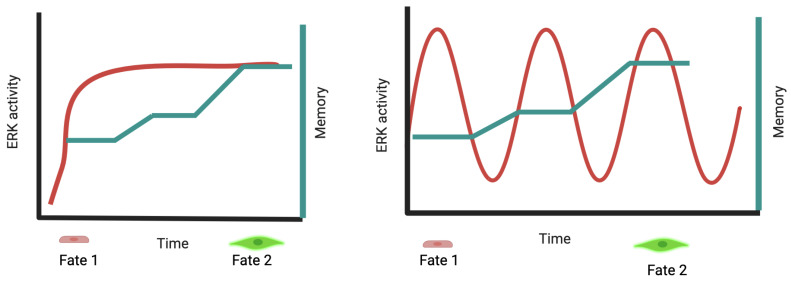
Decoding ERK-signaling intensity by cumulative ERK activation. The red line represents ERK activity. Fate 2 can be attained by a lasting ERK induction or the accumulation of the same signal after several pulses of activation (based on Johnson and Toettcher [52]). This model implies that ERK signaling is somehow remembered in cells receiving short pulses of ERK induction (green line). The nature of the memory signal may depend on the context. In the yeast pheromone pathway, the levels of the CDK inhibitor Far1 reflect an integral of the duration and concentration of past pheromone exposure [53]. Albeck and colleagues earlier proposed this decoding mechanism while studying the induction of the ERK target gene Fra-1. Using a single-cell approach based on fluorescent ERK reporters, they found that both the amplitude and the duration of ERK activity contributed to Fra-1 induction. This graded response apparently contrasts with the Blenis model, where lasting ERK activity was required to induce late-response genes in a switch-like fashion [54]. However, mathematical modeling of the network regulating Fra-1 expression shows that the basal activity of Fra-1 transcription determines whether the network will only respond to long-lasting stimulation as proposed by Blenis or as a signal integrator capable of remembering previous stimulation (adapted from BioRender).

**Figure 6 cancers-16-00095-f006:**
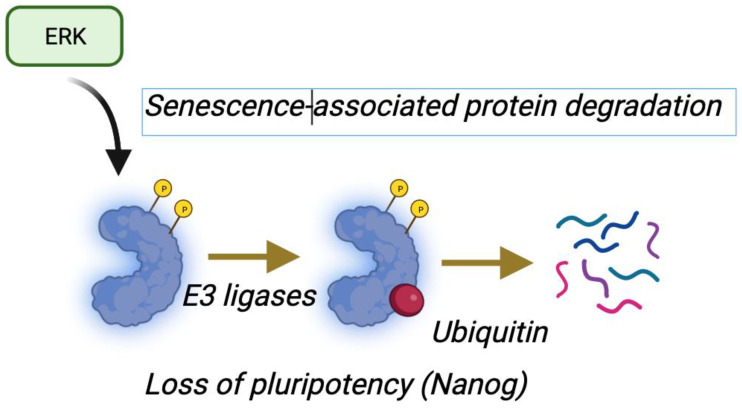
Decoding ERK signal intensity by protein degradation. In this model, high-intensity ERK signals trigger the degradation of many ERK targets by coupling protein phosphorylation to ubiquitin-mediated protein degradation. Prolonged or intense ERK signals are required to increase the stoichiometry of phosphorylation, so subsequent degradation effectively reduces protein levels. This was described in senescent cells as SAPD (senescence-associated protein degradation) [70,75,76] but also in mouse ES cells that lose naïve pluripotency upon ERK activation (adapted from BioRender).

**Figure 7 cancers-16-00095-f007:**
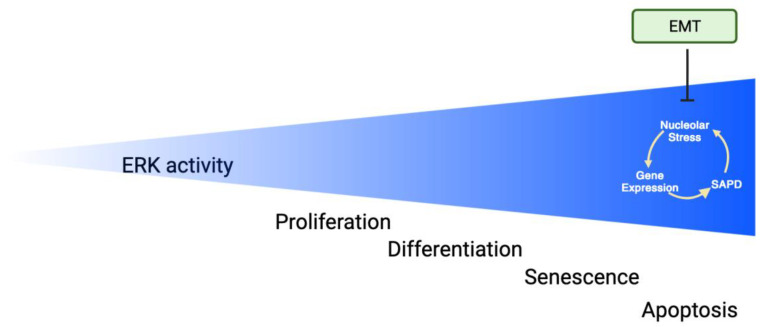
Non-monotonic relationship between ERK activity and cell proliferation. The Goldilocks effect. Proliferation requires moderate ERK activity [10], while differentiation [10], senescence [1,70,71] and apoptosis [79] require stronger signals. Aberrantly high ERK activation triggers protein degradation (SAPD) [70,75,76], changes in gene expression (GE) [77], and nucleolar stress (NoS) [78]. These processes regulate senescence and perhaps apoptosis. EMT reprograms the response to high-intensity ERK signals, acting downstream of ERK to inhibit senescence and apoptosis (Adapted from BioRender).

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
