# Peer review of "Extracellular Signal-Regulated Kinases: One Pathway, Multiple Fates"

_cancers, 2023, doi:10.3390/cancers16010095_

Round 1

Reviewer 1 Report

Comments and Suggestions for Authors

This is an interesting review which provides a helpful summary of modes of ERK activation and appropriate physiological outcome. While various kinetics of activation are described very well, the various locations and translocations of activated ERK1/2 are mentioned but are not described in detail. The review would be more comprehensive when the location issue is described in more detail as well.

Minor points:

Fig. 1: Poor resolution, rather textbook, could be improved, phosphorylation can be schematically shown, why is SOS (son of sevenless) named but the Grb2 protein is not? Abbrevations “A” and “R” in the Legend are not seen in the Figure.

Fig. 2: Poor resolution, could be improved, phosphorylations could be schematically shown.

Fig. 4: What is the difference between frequency and duration in this figure?

Line 111: Frequency is inverse duration of the pulse – two names for the same...

Fig. 6: Red and yellow curve in the left part. Red (ubiquitin) and yellow (phosphorylation) modification in the right part. But colors do not correspond with each other between the two panels. Change colors.

Fig. 8: Align text and symbols.

A Figure summarizing part 6 EMT (lines 255-303) would be helpful.

The extensive part 2 “... survival” and the small part 7 “Apoptosis” should be merged.

There is a subheading 2.1 but no subheading 2.2.

Conclusions: Should provide some general take home massages. CDK4 story seems misplaced here.

Author Response

We express our gratitude to both reviewers for their insightful comments and precise criticisms. We are indebted to them for the valuable improvements they have contributed to this manuscript. Please find highlighted in yellow in the text the changes we have made to answer your questions.

Reviewer # 1

This is an interesting review which provides a helpful summary of modes of ERK activation and appropriate physiological outcome. While various kinetics of activation are described very well, the various locations and translocations of activated ERK1/2 are mentioned but are not described in detail. The review would be more comprehensive when the location issue is described in more detail as well.

Thank you for the positive feedback. We acknowledge the observation that our review article overlooked the issue of ERK activity localization. In response, we have introduced a new subsection titled "2.1 ERK localization in cell proliferation and survival" (lines 122-138). Additionally, we have supplemented the section on differentiation (lines 202-205) to highlight how cellular localization can influence the balance between differentiation and apoptosis. In total we added and discussed 7 new references on this subject.

Minor points: Fig. 1: Poor resolution, rather textbook, could be improved, phosphorylation can be schematically shown, why is SOS (son of sevenless) named but the Grb2 protein is not? Abbrevations “A” and “R” in the Legend are not seen in the Figure.

We improved the resolution, added GRB2, MEK and ERK phosphorylation and expanded the legend. We also removed A and R from the legend.

Fig. 2: Poor resolution, could be improved, phosphorylations could be schematically shown.

We improved the resolution, added phosphorylations to IEG and expanded the legend.

Fig. 4: What is the difference between frequency and duration in this figure? Line 111: Frequency is inverse duration of the pulse – two names for the same...

Yes, the reviewer is right, frequency is simply the inverse of the duration of pulses. Frequency describes how rapidly oscillations occur per unit of time, while duration refers to the time it takes for a single oscillation to complete. Hence, the shorter the duration the higher the frequency. We expanded the legend of the figure to clarify this concept.

Fig. 6: Red and yellow curve in the left part. Red (ubiquitin) and yellow (phosphorylation) modification in the right part. But colors do not correspond with each other between the two panels. Change colors.

We opted for removing the left part and do not speculate about the relationship between the duration of the ERK signal and SAPD. We kept the same colors since there is no longer any color mismatch now.

Fig. 8: Align text and symbols.

 We redid the figure and address this issue.

A Figure summarizing part 6 EMT (lines 255-303) would be helpful.

We added EMT to the model in figure 8.

The extensive part 2 “... survival” and the small part 7 “Apoptosis” should be merged.

We prefer not to merge survival which is a pro-cancer event with apoptosis, which opposes cancer.

There is a subheading 2.1 but no subheading 2.2.

Now there is 2.2 since we added a new sub-session for the localization of ERK activity.

Conclusions: Should provide some general take home massages. CDK4 story seems misplaced here.

We removed the CDK4 work and we expanded the conclusions to give clear take home messages and mention current gaps in the field.

Reviewer 2 Report

Comments and Suggestions for Authors

Review of Xavier DescheÌ‚nes-Simard et al for ‘Cancers’

This review deals with the different phenotypes (or fates) that can be imposed by MAP kinases of the Erk family. The issue is important as the mechanisms through which ERKs affect cells so differently (may cause on one hand proliferation and even oncogenic transformation, and on the other hand differentiation or senescence) are enigmatic. The review is systematic, divided to chapters, based on the phenotypes, and covers the literature relevant to the particular phenotype. As such it is informative and valuable. It also attempts to provide some explanation for the enigma, mostly on the basis of suggestion in the literature (transient vs. sustained activity; pulses of activity vs. constant..), but does not give rise to novel ideas or analyzing deeply current notions. In this regard, the figures showing waves of activity are somewhat naïve and not very helpful. They may be more useful with a more detailed explanation in the legends and perhaps with examples of specific experimental systems or physiological situations in which each mode of activity occurs.

Some specific comments:

The paragraph in lines 73-80 is not read well, perhaps some of the sentences are not in the right order.

The ‘Blenis model’ was not clear to me, and its description is somewhat vague. Does it really explain why in some cases the activated proteins are not stabilized (i.e., why the second step does not occur?)? The authors are advised to either explain this better or to remove the vague idea altogether.

In the ‘proliferation and survival’ section the author must mention that mutations in Erk themselves are being constantly identified in cancer patients. Some seem to play a causative role as they oncogenically transform cells in culture (i.e., Erk1(R84S); Erk1(R84H)).

The paragraph in lines 167-188 should be expanded a bit to explain exactly how Erk may promote or suppress stemness. As it is put now it is confusing. Line 167: “Active ERK1/2 inhibits the `naïve’ pluripotent state “. Line 188: “ERK1/2 activation through FGF promotes 188 the primed pluripotent state .

Figures:

All color parts are vague and very unclear, at least on my screen. Legend to Fig. 1 must be more detailed. Sos is clearly not an ‘adapter protein’ it is a mistake.

As said above many of the figures are either naïve or too simples, unless an explanation and examples are provided.

Author Response

We express our gratitude to both reviewers for their insightful comments and precise criticisms. We are indebted to them for the valuable improvements they have contributed to this manuscript. Please find highlighted in yellow in the main text the changes we have made to answer your questions.

Reviewer #2

This review deals with the different phenotypes (or fates) that can be imposed by MAP kinases of the Erk family. The issue is important as the mechanisms through which ERKs affect cells so differently (may cause on one hand proliferation and even oncogenic transformation, and on the other hand differentiation or senescence) are enigmatic. The review is systematic, divided to chapters, based on the phenotypes, and covers the literature relevant to the particular phenotype. As such it is informative and valuable. It also attempts to provide some explanation for the enigma, mostly on the basis of suggestion in the literature (transient vs. sustained activity; pulses of activity vs. constant..), but does not give rise to novel ideas or analyzing deeply current notions. In this regard, the figures showing waves of activity are somewhat naïve and not very helpful. They may be more useful with a more detailed explanation in the legends and perhaps with examples of specific experimental systems or physiological situations in which each mode of activity occurs.

The critic accurately points out concerns regarding the figures, prompting us to extensively revise them by incorporating additional text and references into the legends. We acknowledge the reviewer's observation regarding the absence of a comprehensive analysis of current concepts in the ERK signaling field. It is essential to note, however, that our primary objective was to emphasize the apparent paradox surrounding the dual pro and anti-tumor effects of ERK activation. This is an area in which we have made original contributions, and we believe that this specific review will be particularly valuable, especially for those unfamiliar with the field who may still perceive ERK in cancer as solely oncogenic.

To illustrate, a major recent review on ERK by Lavoie, H., Gagnon, J., & Therrien, M. (Nat Rev Mol Cell Biol 21, 607–632, 2020) does not address the concept we present here, namely, that hyperactivation of ERK can drive senescence or apoptosis. Additionally, the review overlooks the role of epithelial-mesenchymal transition (EMT) downstream of ERK, which suppresses these responses. In response, we have integrated into our introduction the following statement:

"Here, we concentrate on elucidating potential mechanisms that account for the diverse and occasionally contradictory cellular phenotypes resulting from ERK activation. The complex molecular intricacies governing the ERK pathway have been recently examined (Lavoie et al 2020)."

Some specific comments:

The paragraph in lines 73-80 is not read well, perhaps some of the sentences are not in the right order.

We rewrote and reorganized the paragraph.

The ‘Blenis model’ was not clear to me, and its description is somewhat vague. Does it really explain why in some cases the activated proteins are not stabilized (i.e., why the second step does not occur?)? The authors are advised to either explain this better or to remove the vague idea altogether.

We rewrote this section and the figure legend associated to the Blenis mode (lanes 79-85). The Blenis model does not explain the behavior of every late acting gene induced by ERK, but it was the first to propose a molecular mechanism to explain how lasting signals can induce genes that are not activated by transient signals. We included references to recent variations on this model including the work showing that late acting genes are also stabilized at the mRNA level and the work of Toettcher lab showing that IEG in general respond both to ERK dependent transcriptional control and a more complex and context dependent post-transcriptional control (lanes 91-94).

In the ‘proliferation and survival’ section the author must mention that mutations in Erk themselves are being constantly identified in cancer patients. Some seem to play a causative role as they oncogenically transform cells in culture (i.e., Erk1(R84S); Erk1(R84H)).

Thanks for this suggestion. We included comments and refs. on some of these mutants and discussed their role alone the general model of our review that highlights the relationship between ERK activity and outcome (lanes 98-103).

The paragraph in lines 167-188 should be expanded a bit to explain exactly how Erk may promote or suppress stemness. As it is put now it is confusing. Line 167: “Active ERK1/2 inhibits the `naïve’ pluripotent state “. Line 188: “ERK1/2 activation through FGF promotes 188 the primed pluripotent state” .

We have now explained the differences between the naïve and the primed pluripotent stem cell states that should clarify the potential confusion.

Figures:

All color parts are vague and very unclear, at least on my screen. Legend to Fig. 1 must be more detailed. Sos is clearly not an ‘adapter protein’ it is a mistake. As said above many of the figures are either naïve or too simples, unless an explanation and examples are provided.

We remake all figures at higher resolution, expanded all legends and added references to them.

Round 2

Reviewer 1 Report

Comments and Suggestions for Authors

My former points were taken. May be, Figures 7 and 8 could be merged.

Author Response

Thanks again for useful suggestions. We merged Fig 7 and 8 as recommended.